# Defining the Protease and Protease Inhibitor (P/PI) Proteomes of Healthy and Diseased Human Skin by Modified Systematic Review

**DOI:** 10.3390/biom12030475

**Published:** 2022-03-20

**Authors:** Callum Stewart-McGuinness, Christopher I. Platt, Matiss Ozols, Brian Goh, Tamara W. Griffiths, Michael J. Sherratt

**Affiliations:** 1Division of Cell Matrix Biology & Regenerative Medicine, Faculty of Biology, Medicine and Health, School of Biological Sciences, The University of Manchester, Manchester M13 9PT, UK; callum.stewart-mcguinness@postgrad.manchester.ac.uk (C.S.-M.); matiss.ozols@manchester.ac.uk (M.O.); brian.j.s.goh@gmail.com (B.G.); michael.j.sherratt@manchester.ac.uk (M.J.S.); 2Department of Human Genetics, Wellcome Sanger Institute, Genome Campus, Hinxton CB10 1SA, UK; 3British Heart Foundation Centre of Research Excellence, University of Cambridge, Cambridge CB2 0QQ, UK; 4Centre for Dermatology Research, The University of Manchester & Salford Royal NHS Foundation Trust, Manchester Academic Health Science Centre, Manchester M13 9PL, UK; tamara.griffiths@manchester.ac.uk

**Keywords:** proteome, protease, protease inhibitor, skin, human

## Abstract

Proteases and protease inhibitors (P/PIs) are involved in many biological processes in human skin, yet often only specific families or related groups of P/PIs are investigated. Proteomics approaches, such as mass spectrometry, can define proteome signatures (including P/PIs) in tissues; however, they struggle to detect low-abundance proteins. To overcome these issues, we aimed to produce a comprehensive proteome of all P/PIs present in normal and diseased human skin, in vivo, by carrying out a modified systematic review using a list of P/PIs from MEROPS and combining this with key search terms in Web of Science. Resulting articles were manually reviewed against inclusion/exclusion criteria and a dataset constructed. This study identified 111 proteases and 77 protease inhibitors in human skin, comprising the serine, metallo-, cysteine and aspartic acid catalytic families of proteases. P/PIs showing no evidence of catalytic activity or protease inhibition, were designated non-peptidase homologs (NPH), and no reported protease inhibitory activity (NRPIA), respectively. MMP9 and TIMP1 were the most frequently published P/PIs and were reported in normal skin and most skin disease groups. Normal skin and diseased skin showed significant overlap with respect to P/PI profile; however, MMP23 was identified in several skin disease groups, but was absent in normal skin. The catalytic profile of P/PIs in wounds, scars and solar elastosis was distinct from normal skin, suggesting that a different group of P/PIs is responsible for disease progression. In conclusion, this study uses a novel approach to provide a comprehensive inventory of P/PIs in normal and diseased human skin reported in our database. The database may be used to determine either which P/PIs are present in specific diseases or which diseases individual P/PIs may influence.

## 1. Introduction

Proteases and protease inhibitors (P/PI) play key roles in a plethora of biological processes such as embryogenesis, organ growth, ageing, cancer, inflammation and wound healing [1]. Extracellular proteases, such as matrix metalloproteases (MMPs) modify the extracellular environment by degrading the extracellular matrix (ECM) and by processing secreted cytokines and growth factors, while intracellular proteases, such as caspases, initiate apoptosis. In the skin, proteases mediate tissue homeostasis via regulation of keratinocyte desquamation in the epidermis and the wound healing response in the dermis and epidermis following injury [2]. The crucial role of proteases in normal skin functions is highlighted in vivo using knockout mice in which specific protease genes are deleted. Mice negative for MMP-8, -9 and -13 exhibit delayed wound healing, brought about by an aberrant inflammatory response, delayed re-epithelialisation and wound contraction [3,4]. Granzyme-B is upregulated in injured skin and chronic skin diseases, such as pemphigoid and atopic dermatitis. Because of its ability to cleave anchoring proteins of the dermal-epidermal junction (DEJ), in addition to E-cadherin and filaggrin, studies propose that granzyme-B exacerbates dermal-epidermal separation in pemphigoid and epidermal barrier disruption in dermatitis [5]. Under normal circumstances the expression and activity of proteases in skin and other organs is tightly regulated to prevent excessive tissue destruction which could lead to chronic disease and loss of tissue function [6]. In healthy skin, the mechanisms that mediate protease activity occur at multiple levels, from the control of protease gene transcription [7] through to secretion of proteases as inactive zymogens [8], and the selective activation [9] and suppression of activity by specific inhibitors [10]. However, failure of these mechanisms, through genetic or acquired causes, may contribute to excessive protease activity and the development of skin conditions such as Netherton syndrome [11], rosacea [12], chronic wounds [13] and photoageing [14]. For example, the skin inflammation and scaling characteristic of Netherton syndrome is caused by loss-of-function mutations in *SPINK5*, which encodes the epidermal serine protease inhibitor LEKTI. Absence of LEKTI increases the level of active kallikreins, leading to the cleavage of corneodesmosomes, amongst other epidermal proteins, and subsequent detachment of the stratum corneum [15]. Given the clinical importance of protease activity to skin health it is important to be able to define the P/PI proteomes of healthy and diseased human skin.

Whilst there are many excellent studies showing the expression of P/PIs in human skin, these studies generally focus on a small number of related P/PIs and do not take into account the global P/PI network that may impact protease activity and inhibition [16,17]. Defining the proteome of P/PIs in human skin, therefore, is a crucial first step towards highlighting novel interactions between individual P/PIs and between catalytic groups. To our knowledge, there is currently no comprehensive proteome of proteases and protease inhibitors in human skin. Although unbiased “omics” approaches, such as mass spectrometry (MS), can assess global protein levels (including P/PIs) in tissues [18,19], these approaches may not be the most appropriate platform for identifying low-abundance proteins in a tissue sample [20]. Complimentary antibody-based detection methods can achieve a higher sensitivity but are limited to screening for selected tissue components. With regard to defining the proteases themselves, reviews of individual protease families are highly informative [21], but the MEROPS database provides a comprehensive classification of P/PIs across multiple organisms [22].

We have previously used a modified systematic review approach to help define the healthy human skin proteome by collating thousands of studies [18]. Here, we use a similar methodology to generate a comprehensive inventory of P/PIs (as defined by MEROPS) in healthy and diseased human skin. By using a combination of electronic literature searching and manual review, we have identified a core signature of P/PIs that have been previously reported in healthy human skin. In addition, we compare the healthy skin proteome with discrete P/PI proteomes found in skin diseases such as malignant tumours, inflammatory skin diseases, wounds and solar elastosis, and autoimmune connective tissue disorders. By using a modified systematic review, we have synthesised findings from multiple studies (using both proteomics and antibody-based detection methods) to define the current P/PI proteome of human skin as reported in the peer-reviewed literature. This resource may be used to either identify the network of P/PIs which operate in a disease category/specific disease or to identify the body of disorders in which individual P/PIs may play important roles.

## 2. Materials and Methods

Figure 1 provides an overview of the systematic methodology and review process. This approach uses Web of Science to search all known human P/PIs (from MEROPS) against advanced search terms relating to human skin. The resulting output is a list of articles that is manually reviewed against defined inclusion/exclusion criteria to produce a list of P/PIs present in human skin. 

A key resource in protease research is the MEROPS online peptidase database (www.ebi.ac.uk/merops/cgi-bin/ansearch) (accessed on 17 March 2022), which uses a hierarchical system to classify thousands of prokaryotic and eukaryotic P/PIs into groups [23]. Proteases with a homologous tertiary structure are grouped into *clans*; each clan contains one or more *families* composed of proteases with a homologous amino acid sequence, and families contain *protein-species* categorised according to shared characteristics including inhibitor interactions, substrate preference and phylogeny. Protease clans are named according to their catalytic mechanism: aspartic proteases, glutamic proteases, metalloproteases, serine proteases, threonine proteases and cysteine proteases. Protease inhibitors are classified according to the same system as proteases in MEROPS [24]; however, a single polypeptide chain may contain multiple inhibitory domains. In this case, the protease inhibitor is identified according to each individual inhibitor domain, known as the *inhibitor unit*. In this study, proteases in human skin are reported as clan (catalytic mechanism). 

To define the text corpus of the literature, search methods were adapted from [18]. A total of 986 unique Uniprot accession numbers relating to human proteases and protease inhibitors were collected from the MEROPS online protease database (release 12.3) on 15 July 2020. Accession numbers for each P/PI were validated against Uniprot/Swiss-Prot to confirm that the entries from MEROPS are manually annotated and reviewed by Uniprot curators (Figure 1a). For each accession number the protein name (and alternative names) was retrieved from Uniprot/Swiss-Prot and MEROPS. For each P/PI, a python script (https://github.com/maxozo/Protease_Proteome) (accessed on 17 March 2022) combined generic search terms related to human skin with all possible combinations of protein name(s) and accession numbers to produce a bespoke search string. 

The Web of Science (WOS) core collection was then queried on 16 July 2020 with each search string using the following data constraints: articles written in English only and published between 1990 and 2020. 464 P/PIs relating to human skin were retrieved with at least one article. Abstracts relating to each P/PI were downloaded and curated to be suitable for manual review (Figure 1a). If >2000 articles were retrieved for a P/PI, then articles were ordered by relevance and the first 2000 articles downloaded for manual review. 

All articles resulting from the electronic search (DOI or WOS accession numbers, and Pubmed ID codes) were manually reviewed by two reviewers (Figure 1b). Articles for each P/PI were included in the final dataset if they were present in normal human skin or diseased skin in vivo (including skin secretions such as wound fluid and sweat; normal skin was defined as a skin biopsy from individuals with no diagnosed skin condition) and the P/PI was a translated protein that was detected by proteomic techniques (mass spectrometry) or immunological techniques (ELISA, immunochemistry, Western blotting or specific enzyme activity (e.g., zymography). For normal skin, articles were excluded if the P/PI was present in “normal-appearing/non-lesional skin” from patients with skin disease (e.g., skin adjacent to psoriasis plaques [25]). For diseased skin, articles were excluded if the disease/condition was induced experimentally (e.g., UV irradiation, wounds). Articles were excluded if the P/PI was found in primary human cells, organ cultured human skin in vitro, in the skin of an animal model, present as an mRNA species, or if the P/PI protein was detected using a non-specific histological stain (e.g., cholinesterase) or a non-specific primary antibody (e.g., pan-cathepsin) [26], which would preclude differentiation between P/PI family members (Figure 1b). 

Proteases were categorised according to specific parameters described in MEROPS: catalytic type (aspartic proteases, metalloproteases, serine proteases, cysteine proteases). Protease inhibitors were categorised according to evidence of protease inhibition in vitro and the catalytic family of protease inhibited. Skin diseases were assigned to 11 groups, according to skin pathology and pathophysiology (Appendix A), by a clinical dermatologist. The groups were: malignant tumours; benign tumours; inflammatory dermatoses; autoimmune connective tissue disorders; autoimmune bullous diseases; wounds, scars and solar elastosis; infections; dry skin, ichthyosis and genodermatoses; pigmentation disorders; granulomatous diseases; infiltrative diseases. Using the curated dataset, we have compared the P/PI proteomes of normal with diseased skin; normal with broad disease categories and, in the database facilitate, comparisons between normal skin and individual diseases (Figure 1c).

## 3. Results

Following manual review, the final dataset included 111 proteases and 77 protease inhibitors (Appendix A). Because MEROPS classifies Cathepsin-L2, -L, -S and –K as both proteases and protease inhibitors, these four proteins were included as both proteases and protease inhibitors. Four catalytic protease families were present in skin: serine, metallo- (metalloprotease), cysteine and aspartic. Proteases belonging to threonine and glutamic acid families were absent (Figure 2a). The dataset also contained non-peptidase homologs (NPH); proteases described by MEROPS as containing a known sequence, which can be placed in a peptidase family, but which lack one or more of the expected catalytic residues.

To determine whether any catalytic groups were over-/under-represented in skin, the total number of proteases in skin was compared with all known human proteases in MEROPS. Human skin showed the same pattern as MEROPS with respect to the relative proportion of serine > metallo- > cysteine > NPH > aspartic protease families (Figure 2a). In each catalytic group, the proportion of proteases in skin was <25% of the total number of human proteases, suggesting that a large proportion of human protease proteins have not been identified or are absent in skin (Figure 2a). The greatest number of published articles in human skin were related to metalloproteases (Figure 2b), and the top 10 most frequently identified proteases in human skin were MMP9 followed by MMP2 > MMP1 > chymase > uPA > cathepsin-D > MMP3 > cathepsin-L > TGM1 > neutrophil elastase > MMP7 > BRCA1-associated protein-1 > tPA (identified in 10 or more articles) (Appendix A).

Protease inhibitors were grouped according to the catalytic class of protease they inhibited (Figure 2c). Most protease inhibitors in skin were shown to inhibit a single catalytic class of protease; however, alpha-2-macroglobulin, alpha-2-macroglobulin-like protein, cystatin-C, SERPIN-A5, SERPIN-B3 and SERPIN-B4 inhibited proteases across multiple families. For each protease inhibitor, published evidence of in vitro protease inhibition was used to distinguish between biochemically active protease inhibitors and inhibitors with no reported protease inhibitory activity (NRPIA). The largest group of inhibitors was NRPIA, containing 49.4% of protease inhibitors found in skin, followed by serine > cysteine > metallo- > and aspartic. The proportion of protease inhibitors targeting a specific catalytic group in skin followed the same trend as MEROPS (Figure 2c); however, NRPIAs appeared to be under-represented in skin. For serine, cysteine, metallo- and aspartic families, the proportion of protease inhibitors in skin was >50% of the total number of biologically active human protease inhibitors (Figure 2c). For protease inhibitors, the highest number of published articles related to NRPIA (Figure 2d), and the most frequently published protease inhibitors in skin (identified in 10 or more articles) were complement-C3 > TIMP1 > SPINK5 > C-type lectin domain family four member C > Von Willebrand factor > TIMP2 > cathepsin-L (Appendix A).

Because P/PIs play a significant role in normal skin homeostasis and in disease progression, the dataset was next interrogated to determine the profile of individual P/PIs in normal skin and diseased skin. Only proteases and protease inhibitors with known proteolytic activity and evidence of protease inhibition, respectively, were analysed. The numbers of proteases identified in normal and diseased skin were 84 and 83, respectively (Figure 3a), and the numbers of protease inhibitors identified in normal and diseased skin were 31 and 34, respectively (Figure 3b). There was a substantial overlap in P/PIs identified in both normal and diseased skin, characterised by the presence of cathepsins, caspases, matrix metalloproteases and kallikreins (Figure 3a), while cathepsins, SERPINs and TIMPs represented the protease inhibitors (Figure 3b). In general, the number of articles describing P/PIs in skin disease far exceeded articles describing P/PIs in normal skin (Appendix A).

The dataset was next investigated to determine whether the P/PI profile changed according to the type of skin disease. Individual skin diseases were allocated to a skin disease group by a clinical dermatologist (author, T.W.G.). For cutis laxa and acne vulgaris, an individual disease group was not clearly identified. In this case, these diseases/conditions were allocated to more than one disease group (Appendix A). The number of P/PIs present in skin showed an apparent difference according to the disease group (Figure 4a); however, this may be due to publication bias as the number of published articles associated with each disease group follows the same trend (Figure 4b). Cathepsin-D and cathepsin-B; MMP9; CMA1, uPA and tPA, as representatives of the aspartic, cysteine, metallo- and serine protease families, respectively, were the most frequently identified proteases across all disease groups (including normal skin). Protease inhibitors A2M, cathepsin-L, TIMP1 and PAI2, representing inhibitors of the aspartic, cysteine, metallo- and serine protease families, respectively, were the most frequently identified across all disease groups (including normal skin). It is of interest that MMP23, which was absent in normal skin, was present in four skin disease groups.

To determine whether the catalytic profile of P/PIs changed according to the type of skin disease, inflammatory dermatoses, malignant tumours and wounds, scars and solar elastosis were chosen because these 3 groups contained the greatest number of P/PIs in skin. In normal skin, the predominant family was serine protease, followed by cysteine > metallo- > aspartic (Figure 5a). Inflammatory dermatoses followed the same trend as normal skin, albeit with fewer proteases, and included caspase-5, MMP15, granzyme-H and tryptase alpha, which were absent in normal skin (Figure 5b). The predominant family in malignant tumours was metalloprotease, followed by serine > cysteine > aspartic (Figure 5c). Proteases present in malignant tumours but not in normal skin included: PGC, BAP1, caspase-8, F13A1, MMP11, MMP23, MMP25, MMP26 and granzyme-A. The predominant catalytic family in wounds, scars and solar elastosis was the metalloprotease, followed by serine > cysteine > aspartic (Figure 5d). Proteases present in wounds but not in normal skin included: ADAM15, IDE, MMP23 and MMP28. For protease inhibitors, the proportion of inhibitors that targeted a particular catalytic family showed the same pattern in normal skin, inflammatory dermatoses, and malignant tumours: serine > cysteine > metallo- (Figure 5e–g). In wounds, scars and solar elastosis, the proportion of protease inhibitors was serine > metallo- > cysteine (Figure 5h). Inhibitors targeting aspartic proteases were absent in all four groups. Individual protease inhibitors that were absent in normal skin but present in the other skin disease groups included: SERPINB4 (inflammatory and malignant); BIRC5 and TFPI2 (malignant); and SERPIND1 and TFPI (wounds).

## 4. Discussion

Here, we have established a comprehensive database of proteases and protease inhibitors (P/PIs) in normal and diseased human skin using a modified systematic review. We make the important distinction of only using articles which report P/PI proteins in human skin in vivo, strictly excluding articles which report animal and in vitro models, and which identify P/PI by mRNA expression. In addition, this approach provides a comprehensive review of the literature as all articles identifying a specific P/PI were included in the analysis. This contrasts with other reviews which generally focus on a small number of related P/PIs. In this way, the review provides a useful reference for scientists and clinicians interested in P/PIs in human skin. Another benefit of this approach is the ease of access to data, as we have linked all P/PIs in skin to a specific disease group where applicable. 

This study confirms the presence of the serine, metallo-, cysteine and aspartic acid protease families and their associated inhibitors in human skin. Proteases from the threonine and glutamic acid families are not represented in skin, possibly due to the very low number of these families in humans (three threonine proteases and two glutamic acid proteases in humans). The threonine protease taspase-1 is overexpressed in several human cancer tissues and cell lines and has been shown to regulate spermiogenesis and craniofacial development in mice [27]. Taspase-1 regulates gene transcription in vitro by cleaving and activating the transcription factors mixed-lineage leukemia (MLL) and TFIIA [27]. The glutamic acid proteases tiki1 and tiki2 proteolytically degrade members of the Wnt family of signalling proteins in vertebrates, resulting in Wnt proteins that are unable to bind to their receptors and elicit a cellular response [28] 

P/PIs identified in this study represent a relatively small proportion of all the known P/PIs in humans (as listed in MEROPS), suggesting that skin has a distinct P/PI profile compared with other areas of the body. However, it is possible that some P/PIs listed in MEROPS have not yet been studied in skin, raising the possibility that this core profile of P/PIs identified in skin may expand further. The biologically active proteases MMP9, chymase, uPA and tPA, and the protease inhibitors TIMP1 and cathepsin-L, show a high frequency of publication and are identified in normal skin and across multiple skin disease groups. These proteases and inhibitors directly interact with each other to activate (chymase, uPA, tPA) or inhibit (TIMP1) MMP9, suggesting that regulation of MMP-9 activity is important for the progression of skin disease, as has been reported following UV irradiation of skin [29,30]. The expression of MMP9 in tissues other than skin and its upregulation in pathological processes indicate that MMP9 may be useful as a potential disease biomarker. Higher MMP9 titres have been found in the sputum and saliva of individuals with chronic obstructive pulmonary disease (COPD) and asthma compared to healthy controls [31]; MMP9 is expressed in atherosclerotic plaques where it is believed to play a role in plaque rupture [32]; and MMP9 is involved in several neurological disorders, such as stroke, multiple sclerosis and chronic neuropathic pain [33]. Although aspartic acid proteases constitute a small catalytic group in humans, this study shows that cathepsin-D is present across a wide range of skin disease groups, as well as in normal skin. In addition, cathepsin-D protein is the sixth most frequently published protease in human skin, according to this study. It is of interest that there is a lack of articles surrounding the localisation of BMP1 in normal human skin in vivo, given that BMP1 plays a key role in collagen synthesis. Following the manual review, the role of BMP1 in skin appears to have been elucidated using primarily in vitro or animal models, and so did not satisfy inclusion/exclusion criteria for this study. Conversely, the presence of caspase-2, -4, -6 and -7 in normal, but not diseased skin, may indicate a non-apoptotic role for these proteases in normal skin development and homeostasis [34,35]. The list of all human P/PIs obtained from MEROPS, which was inputted into our electronic search, also contains a sub-group of proteins with no recognizable activity (i.e., protease NPHs, protease inhibitor NRPIA). It is likely that these homologous proteins play other key roles within the skin. For example, the NPH hepatocyte growth factor (HGF) is identified as a protease by MEROPS due to the presence of a serine protease domain; however, HGF’s main mechanism of action in the skin is signalling through its receptor c-Met. In addition, the NRPIA complement C3 is included as a protease inhibitor by MEROPS due to its homology to alpha-2-macroglobulin, but its role in the complement activation cascade is better characterised. 

The number of proteases in normal skin and diseased skin is equivalent, although the profile of individual proteases is distinct. This could suggest that the core proteases which are present in diseased states may be actively translated in normal skin; however, because it is difficult to fully determine the relative abundance of individual proteases in this study, differences in protein expression in normal skin and diseased skin cannot be ruled out. However, MMP15 (MT2-MMP), which is present in the inflammatory dermatoses group but not normal skin, in the current study, is elevated in psoriasis plaques compared with peri-lesional skin [36]. Because MMP15 can cleave a wide variety of ECM components (fibronectin, laminin, tenascin-C, aggrecan and perlecan), in addition to inflammatory (MMP2 and TNF-alpha) and angiogenic (ADAMTS1) factors [37,38], it would be interesting to further explore the role of MMP15 in normal and inflamed skin.

In this study, the protease catalytic profile changes across disease groups, suggesting that individual skin diseases utilise a different proteolytic repertoire as part of their progression. In malignant tumours, there is an increase in the proportion of metalloproteases compared to normal skin, while the proportion of metalloprotease inhibitors in malignant tumours remains approximately the same as in normal skin. This may indicate a potential increase in net metalloprotease activity, which would promote ECM breakdown and invasiveness in tumour growth and development [39]. For wounds, scars and solar elastosis, there is an increase in the proportion of both metalloproteases and metalloprotease inhibitors, compared with normal skin, raising the possibility that increased metalloprotease-mediated ECM degradation during wound healing is controlled by a concomitant increase in metalloprotease inhibition [40]. In addition, a large proportion of caspases (with the exception of caspase-3) are absent in wounds, scars and solar elastosis, compared with normal skin. Caspases regulate apoptosis in normal wounds by controlling the removal of infiltrative cells following the inflammatory response and removing myofibroblasts from the resolving scar. Disruption of apoptosis can lead to pathological wound healing. Reduced caspase expression, related to the persistence of wound myofibroblasts, correlates with excessive fibrosis found in hypertrophic and keloid scars [41]. In contrast, enhanced fibroblast apoptosis, which correlates with increased caspase expression, may lead to delayed granulation tissue formation and healing in chronic wounds [42]. An interesting candidate for further study is MMP23, as it is reported in four disease groups, but not in normal skin, in the current study. MMP23 is primarily expressed in reproductive tissues and cleaves a synthetic peptide in vitro [43]. Although no physiological substrate has yet been identified for MMP23, reports show that MMP23 interacts with and regulates the function of the Kv1.3 potassium channel in transfected COS-7 cells [44].

## 5. Conclusions

By using a modified systematic review, we have generated a thorough and comprehensive proteome of P/PIs in human skin. We have identified the key catalytic protease and inhibitor families in skin and, by organising the data into distinct skin disease categories, we have identified specific P/PI profiles for several skin diseases. We freely store this database for the scientific community and clinical communities benefit on our web platform [45]. This resource aims to be a guide for basic scientists and clinicians interested in exploring the role of P/PIs in human skin homeostasis and disease.

## Figures and Tables

**Figure 1 biomolecules-12-00475-f001:**
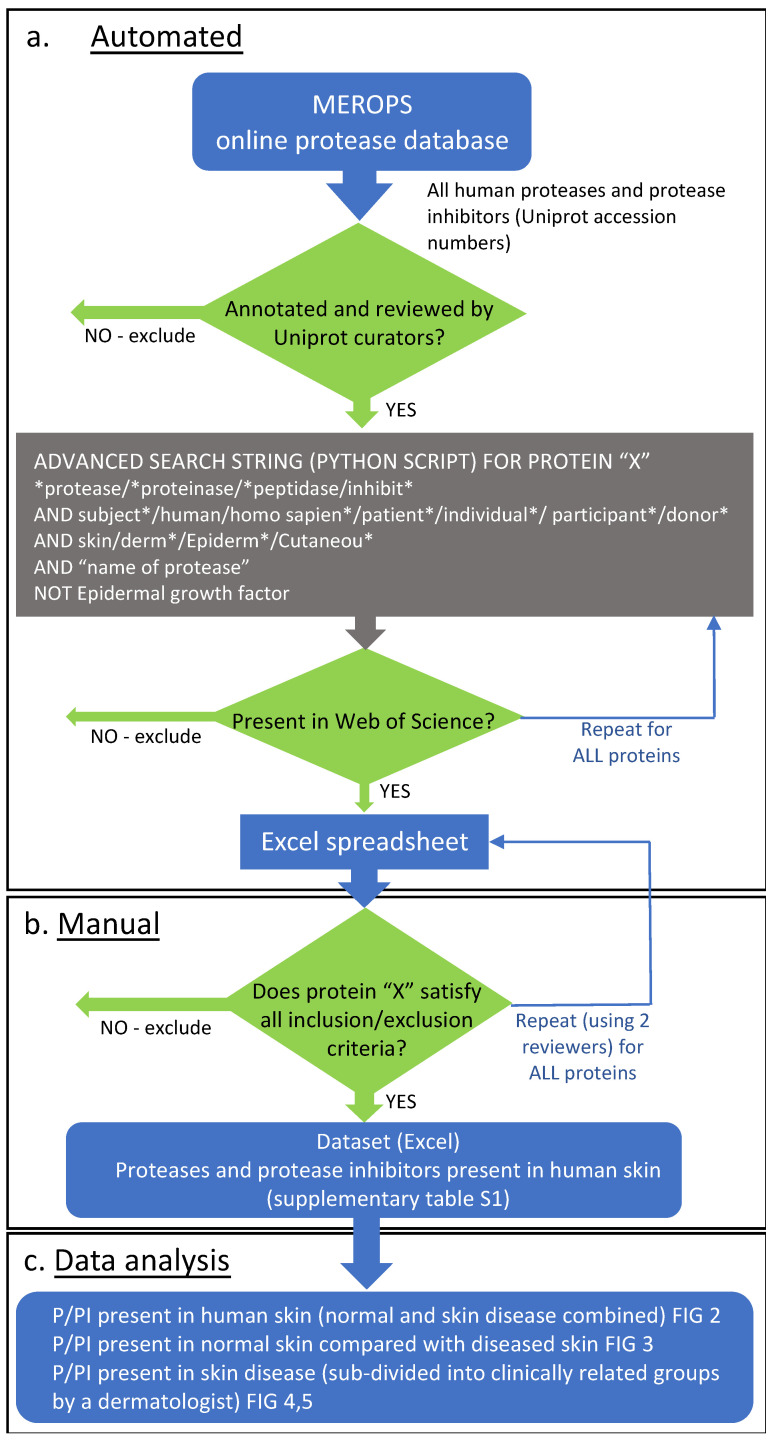
Modified systematic review methodology. (**a**) Automated search: accession numbers for all human proteases and protease inhibitors (P/PIs) from MEROPS were validated against Uniprot/Swiss-Prot to confirm annotation, alternative names and review status; all human P/PI names queried in Web of Science using defined search parameters; data output was a list of articles for each P/PI in an Excel spreadsheet. (**b**) Manual review: articles for each P/PI reviewed and accepted/rejected according to defined inclusion/exclusion criteria; data output was a list of P/PIs (the dataset can be found in Appendix A) present in normal human skin and skin disease. (**c**) Data analysis: presence of P/PIs in all human skin types (normal and disease), presence of P/PIs in normal skin compared with skin disease, presence of P/PIs in skin disease, sub-divided into clinically relevant disease groups.

**Figure 2 biomolecules-12-00475-f002:**
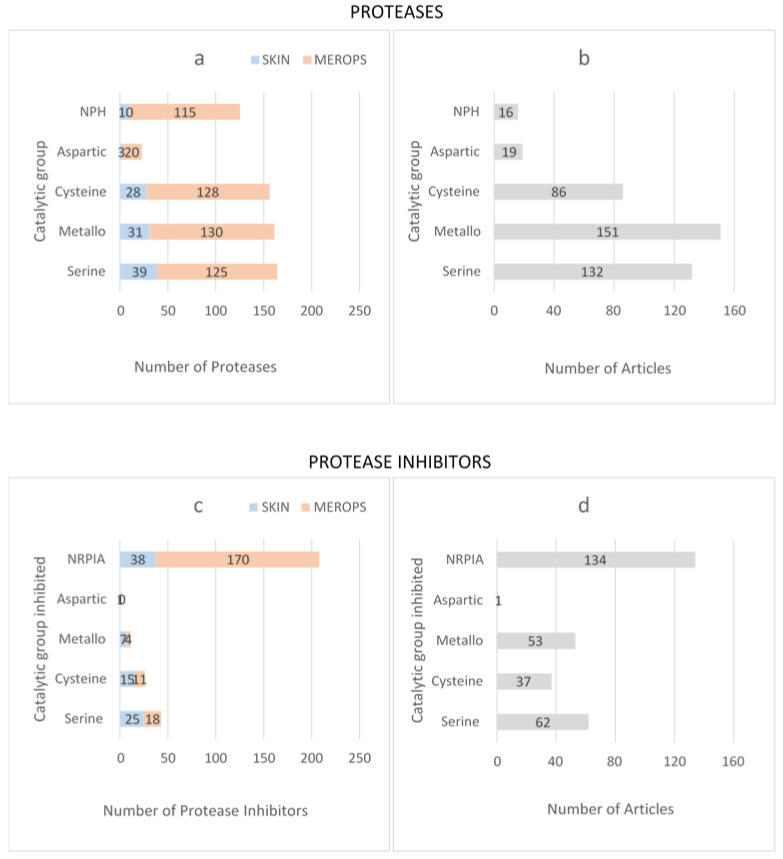
Number of P/PI in human skin, categorised according to catalytic family (protease) or catalytic family that is inhibited (inhibitors). (**a**) Total number of proteases identified in human skin compared to total number of human proteases in MEROPS, grouped according to catalytic family (NPH = non-peptidase homolog. For clarity this group also contains uncharacterised homologs and pseudogenes). (**b**) Total number of published articles grouped according to protease catalytic family. (**c**) Total number of protease inhibitors identified in human skin compared to total number of human protease inhibitors in MEROPS, grouped according to the catalytic family that they inhibit (NRPIA = no reported protease inhibitory activity). (**d**) Total number of published articles grouped according to catalytic families that are inhibited.

**Figure 3 biomolecules-12-00475-f003:**
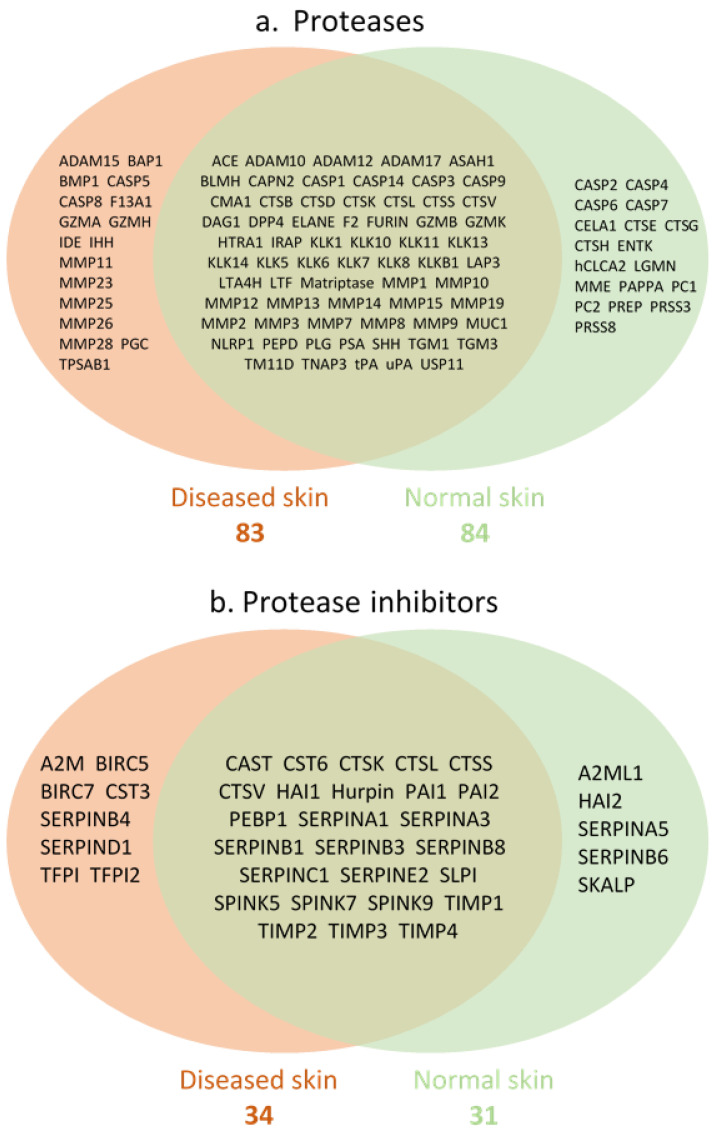
P/PI present in normal and diseased skin. (**a**) Proteases; (**b**) Protease inhibitors.

**Figure 4 biomolecules-12-00475-f004:**
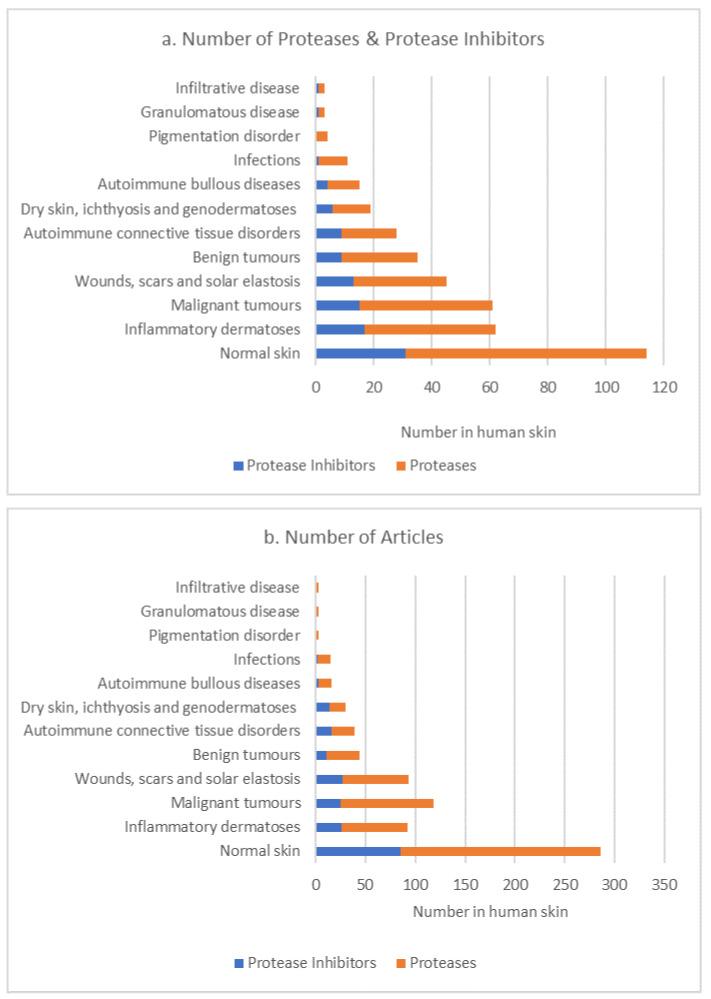
Number of P/PIs in skin disease groups (**a**); number of articles reporting P/PIs in different skin disease groups (**b**).

**Figure 5 biomolecules-12-00475-f005:**
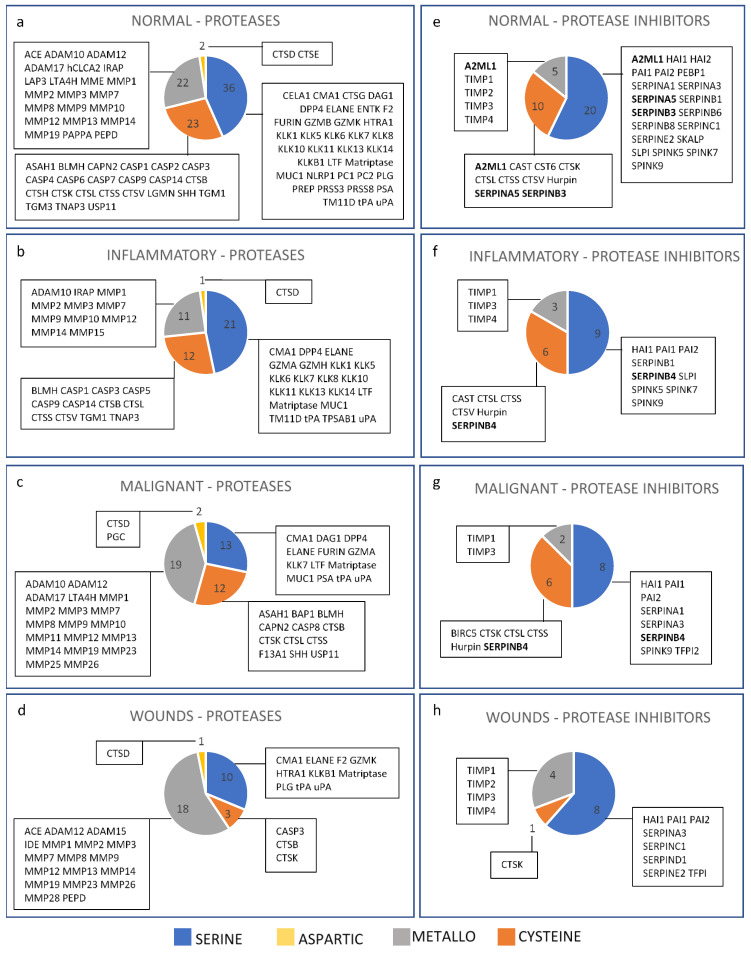
P/PI in normal skin, inflammatory dermatoses, malignant tumours, and wounds, scars and solar elastosis, grouped according to catalytic family (proteases, (**a**–**d**)) or catalytic family inhibited (protease inhibitors, (**e**–**h**)). Protease inhibitors in bold show inhibitors that target more than one catalytic family.

## Data Availability

Online repository for python script used in this study: https://github.com/maxozo/Protease_Proteome; https://www.manchesterproteome.manchester.ac.uk/#/Protease_Proteome.

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
