# Peer review of "Defining the Protease and Protease Inhibitor (P/PI) Proteomes of Healthy and Diseased Human Skin by Modified Systematic Review"

_biomolecules, 2022, doi:10.3390/biom12030475_

Round 1

Reviewer 1 Report

The topic is interesting. However, the following major points should be addressed:

  1. Please briefly highlight the biological significance / rationale of the work presented here.
  2. "Skin Disease" is a very broad term- There is a lot of pathogenic mechanistic variation that exists in several forms of skin disease. Thus, grouping all of them together does not make any sense.
  3. Please bring contextual perspective of proteases and protease inhibitors during specific skin pathology.
  4. Please highlight the role of these Proteases/inhibitors in Skin ECM.

Author Response

  1. Please briefly highlight the biological significance / rationale of the work presented here.

We thank the reviewer for their feedback and have clarified the rationale for this work in the abstract (page 1: lines 31-34):

“In conclusion, this study uses a novel approach to provide a comprehensive inventory of P/PIs in normal and diseased human skin reported in our database https://www.manchesterproteome.manchester.ac.uk/#/Protease_Proteome. The database may be used to determine either which P/PIs are present in specific diseases or which diseases individual P/PIs may influence.”

And also the introduction (page 2: lines 86-92, 109-114):

Whilst there are many excellent studies showing the expression of P/PIs in human skin, these studies generally focus on a small number of related P/PIs and do not take into account the global P/PI network that may impact protease activity and inhibition [12,13]. Defining the proteome of P/PIs in human skin, therefore, is a crucial first step towards highlighting novel interactions between individual P/PIs and between catalytic groups. To our knowledge, there is currently no comprehensive proteome of proteases and protease inhibitors in human skin.

By using a modified systematic review, we have synthesised findings from multiple studies (using both proteomics and antibody-based detection methods) to define the current P/PI proteome of human skin as reported in the peer-reviewed literature. This resource may be used to either identify the network of P/PIs which operate in a disease category/specific disease or to identify the body of disorders in which individual P/PIs may play important roles.

  1. "Skin Disease" is a very broad term - There is a lot of pathogenic mechanistic variation that exists in several forms of skin disease. Thus, grouping all of them together does not make any sense.

We thank the reviewer for the comment and the opportunity to clarify this impotent point. Our intention in comparing normal to “diseased” skin was to determine if any proteases or inhibitors were characteristic of pathology. So, for example, five MMPs (11, 23, 25, 26 and 28) have not yet been reported in healthy skin and so their presence may be indicative of pathology. We have strived to characterize the proteomes at increasing levels of complexity/granularity comparing normal to disease, normal to 11 clinically related disease groups and finally, in the database allowing comparisons between normal and individual diseases. To further clarify this intention for the reader, we have now added a 3rd panel (panel C) to figure 1. This additional panel shows the data analysis process that was carried out for this study: starting with assessing P/PI in skin as a whole (normal and diseased), through to dissecting P/PIs in separate but pathologically related skin diseases/conditions.

Page 4: lines 153-156; and page 5: lines 221-224

“c) Data analysis: presence of P/PIs in all human skin types (normal and disease), presence of P/PIs in normal skin compared with skin disease, presence of P/PIs in skin disease, sub-divided into clinically relevant disease groups.”

“Using the curated dataset we have compared the P/PI proteomes of normal with diseased skin; normal with broad disease categories and, in the database, facilitate comparisons between normal skin and individual diseases (figure 1c).”  

  1. Please bring contextual perspective of proteases and protease inhibitors during specific skin pathology.

In the previous version of the manuscript, we attempted to contextualise P/PIs identified in different diseases (see discussion: MMP-9 regulation and UV, paragraph 3; MMP-15 and psoriasis, paragraph 4; caspases, paragraph 3; MMP-23, paragraph 5). To further contextualise the results, we have additionally discussed the relative contribution of metalloproteases and metalloprotease inhibitors in malignant skin diseases and wounds, and have discussed these results in light of the changing balance of MMP/TIMPs in driving skin disease (Page 13: line 412-420).

“In malignant tumours, there is an increase in the proportion of metalloproteases compared to normal skin, while the proportion of metalloprotease inhibitors in malignant tumours remains approximately the same as normal skin. This may indicate a potential increase in net metalloprotease activity, which would promote ECM breakdown and invasiveness in tumour growth and development [39]. For wounds, scars and solar elastosis, there is an increase in the proportion of both metalloproteases and metalloprotease inhibitors, compared with normal skin, raising the possibility that increased metalloprotease-mediated ECM degradation during wound healing is controlled by a concomitant increase in metalloprotease inhibition”

In addition, we have discussed data from our study, which shows a virtual absence of caspases in the wound healing group compared to normal skin. We consider this in the context of caspase levels/apoptosis in the development of pathological scarring and chronic wounds (Page 13: line 420-428). 

“In addition, a large proportion of caspases (with the exception of caspase-3) are absent in wounds, scars and solar elastosis, compared with normal skin. Caspases regulate apoptosis in normal wounds by controlling the removal of infiltrative cells following the inflammatory response and removing myofibroblasts from the resolving scar. Disruption of apoptosis can lead to pathological wound healing. Reduced caspase expression, related to the persistence of wound myofibroblasts, correlates with excessive fibrosis found in hypertrophic and keloid scars [41]. In contrast, enhanced fibroblast apoptosis, which correlates with increased caspase expression, may lead to delayed granulation tissue formation and healing in chronic wounds”

4. Please highlight the role of these Proteases/inhibitors in Skin ECM.

We have now highlighted the role of key MMPs, granzyme-B (Page 2: line 62-70) and LEKTI (Page 2: lines 78-83) in wound healing, pemphigoid and Netherton syndrome, respectively.

“The crucial role of proteases in normal skin functions is highlighted in vivo using knockout mice in which specific protease genes are deleted. Mice negative for MMP-8, -9 and -13 exhibit delayed wound healing, brought about by an aberrant inflammatory response, delayed re-epithelialisation and wound contraction [3, 4]. Granzyme-B is up-regulated in injured skin and chronic skin diseases, such as pemphigoid and atopic dermatitis. Because of its ability to cleave anchoring proteins of the dermal-epidermal junction (DEJ), in addition to E-cadherin and filaggrin, studies propose that granzyme-B exacerbates dermal-epidermal separation in pemphigoid and epidermal barrier disruption in dermatitis”

“For example, the skin inflammation and scaling characteristic of Netherton Syndrome is caused by loss-of-function mutations in SPINK5, which encodes the epidermal serine protease inhibitor LEKTI. Absence of LEKTI increases the level of active kallikreins leading to cleavage of corneodesmosomes, amongst other epidermal proteins, and subsequent detachment of the stratum corneum”

Reviewer 2 Report

The manuscript by Stewart-McGuinness et al. describes a database collection of protease inhibitors using an advanced algorithm. Although the work is interesting, the significance and merit of the work is not clear. The work could be published in their website or newsletter, but not worthy of publication in a journal. 

Author Response

  1. The manuscript by Stewart-McGuinness et al. describes a database collection of protease inhibitors using an advanced algorithm. Although the work is interesting, the significance and merit of the work is not clear. The work could be published in their website or newsletter, but not worthy of publication in a journal. 

We thank the reviewer for their feedback. The significance of the work relates to this study being, to our knowledge, the first comprehensive proteome of proteases and protease inhibitors in normal and diseased human skin. In this amended version, the authors have provided further clarification for the rationale for this study and have highlighted the potential applications for the database:

“In conclusion, this study uses a novel approach to provide a comprehensive inventory of P/PIs in normal and diseased human skin reported in our database https://www.manchesterproteome.manchester.ac.uk/#/Protease_Proteome. The database may used to determine either which P/PIs are present in specific diseases or which diseases individual P/PIs may influence.”

And also the Introduction (page 2: lines 86-92, 109-114):

“Whilst there are many excellent studies showing the expression of P/PIs in human skin, these studies generally focus on a small number of related P/PIs and do not take into account the global P/PI network that may impact protease activity and inhibition [12,13]. Defining the proteome of P/PIs in human skin, therefore, is a crucial first step towards highlighting novel interactions between individual P/PIs and between catalytic groups. To our knowledge, there is currently no comprehensive proteome of proteases and protease inhibitors in human skin”

“By using a modified systematic review, we have synthesised findings from multiple studies (using both proteomics and antibody-based detection methods) to define the current P/PI proteome of human skin as reported in the peer-reviewed literature. This resource may be used to either identify the network of P/PIs which operate in a disease category/specific disease or to identify the body of disorders in which individual P/PIs may play important roles”

Reviewer 3 Report

The present manuscript submitted by Callum Stewart-McGuinness and coworkers reports the application of a modified systematic review with the aim to create a comprehensive inventory of proteases and protease inhibitors (PIs) in normal and diseased human skin. In particular, the authors have carried out a systematic review using a list of proteases and PIs from the MEROPS database and information form original research articles. After manual curation of the data, the authors claim that they have identified 111 proteases and 77 protease inhibitors in human skin from the serine, metallo-, cysteine and aspartic acid protease families. The MEROPS database is a reference database of proteases. However, this database lacks disease-associated information. In this work, the authors provide a useful workflow for the identification of disease-associated proteases/PIs. Overall, the manuscript is well-presented, and the topic addressed is interesting. However, in my opinion, the data provided is preliminary.

Major comments:

  • The authors claim that they have stablished a comprehensive database of proteases and protease inhibitors. However, they did not provide a link or access to such mentioned database. In my oppinion, the authors should offer a user-friendly freely accesible database with the information derived from the present study.
  • Related to my previous comment, the authors should consider also to include information information regarding other disease-associated tissues.  

Author Response

The authors claim that they have stablished a comprehensive database of proteases and protease inhibitors. However, they did not provide a link or access to such mentioned database.

  1. In my oppinion, the authors should offer a user-friendly freely accesible database with the information derived from the present study.

We thank the reviewer for their feedback.

We have developed an interactive, freely accessible database based upon the findings of this study (website address and access details now included in the data availability section of the amended manuscript – line 485).

The database is available here: https://www.manchesterproteome.manchester.ac.uk/#/Protease_Proteome [Username: Reviewer | password: Proteome2020 – to be made open access upon manuscript acceptance]

Although we have tried to ensure that the database is ready for public use following manuscript revision, there are some additional formatting issues that require attention. In light of this we have modified supplementary table 1 (which contains all P/PI information relating to this study), by adding a search function in the title of each column so users may search for a particular P/PI, or a specific skin disease.

  1. Related to my previous comment, the authors should consider also to include information information regarding other disease-associated tissues.

As a representative protease, and due to its prominence in this study, the authors have further discussed the expression of MMP-9 in other tissue diseases (line 331-338).

“The expression of MMP9 in tissues other than skin and its upregulation in pathological processes indicate that MMP9 may be useful as a potential disease biomarker. Higher MMP9 titres have been found in the sputum and saliva of individuals with chronic obstructive pulmonary disease (COPD) and asthma compared to healthy controls [31]; MMP9 is expressed in atherosclerotic plaques where it is believed to play a role in plaque rupture [32]; and MMP9 is involved in several neurological disorders such as stroke, multiple sclerosis and chronic neuropathic pain”

Round 2

Reviewer 1 Report

Accepted

Reviewer 2 Report

Some of the issues were addressed.